# Splicing Measurement and Compensation of Straightness Errors for Ultra-Precision Guideways

**DOI:** 10.3390/mi14091670

**Published:** 2023-08-26

**Authors:** Lian Zhou, Nan Zheng, Jie Li, Zhigang Yuan, Jian Wang, Fei Fang, Qiao Xu

**Affiliations:** Research Center of Laser Fusion, China Academy of Engineering Physics, Chengdu 610093, China; nanceerie@163.com (N.Z.); cdjennylj@sina.com (J.L.); yuanzhigang23@163.com (Z.Y.); wj7130@sina.com (J.W.); fanfei65790299@163.com (F.F.); xuqiao@vip.sina.com (Q.X.)

**Keywords:** straightness error, ultra-precision guideway, splitting measurement, error compensation, flatness error

## Abstract

The straightness error of guideways is one of the key indicators of an ultra-precision machine, which plays an important role in the machining accuracy of a workpiece. In order to measure the straightness error of a long-distance ultra-precision guideway accurately, a splicing measurement for the straightness error of a guideway using a high-precision flat mirror and displacement sensor was proposed in this paper, and the data splicing processing algorithm based on coordinate transformation was studied. Then, comparative experiments on a splicing measurement and direct measurement of the straightness error were carried out on a hydrostatic guideway grinder. The maximum difference between the two measurements was 0.3 μm, which was far less than the straightness error of 5.8 μm. The experiment demonstrated the correctness of the proposed splicing measurement method and data processing algorithm. To suppress the influence of the straightness error on machining accuracy, a straightness error compensation algorithm based on error rotation transformation and vertical axis position correction was proposed, and the grinding experiment of a plane optics with a size of 1400 mm × 500 mm was carried out. Without error compensation grinding, the flatness error of the element was 7.54 μm. After error compensation grinding, the flatness error was significantly reduced to 2.98 μm, which was less than the straightness errors of the guideways. These results demonstrated that the straightness error of the grinding machine had been well suppressed.

## 1. Introduction

With the continuous development of industrial technology, long-distance and high-precision hydrostatic linear guideways are increasingly being applied in advanced manufacturing fields, such as large-diameter ultra-precision turning machines, ultra-precision grinder, coordinate measuring systems [1,2], etc. The straightness error is one of the most important geometric errors of the guideway, which not only affects the machining accuracy, but also the reliability and stability of the equipment during long-term operation [3]. In the process of equipment manufacturing, there are many methods which can effectively reduce the straightness error of the guideway, such as improving the flatness of the guideway working surface [4], optimizing the fluid parameters in the slider [5] and reducing hydraulic oil temperature fluctuations [6]. During the equipment debugging stage, by accurately measuring the straightness error of the guideway and compensating it in the CNC system, the straightness error of the guideway can be further reduced [7]. For machine tools such as ultra-precision lathes and grinders, the straightness error of the guideway will be copied onto the surface of the component, which affects the machining accuracy [8]. Therefore, measuring the straightness error accurately, analyzing the influence of the straightness error on machining accuracy, establishing a compensation algorithm for the straightness error and error compensating during the machining process are effective means to improve the machining accuracy of components [9].

There are many methods for measuring the straightness error of a guideway. Ahmed Elmelegy measured the geometric error of a machine with an autocollimator and laser interferometer, and analyzed the influence of different methods on the measurement results [10]. Chen proposed a simultaneous measurement of the straightness error and its position using a modified Wollaston-prism-sensing homodyne interferometer and presented the optical configuration and the measurement principle [11]. Chen developed a system consisting of a novel one-dimension probe and a ball array to quickly measure the geometric error of a linear axis from the ball center deviations in three dimensions [12]. Vladas Vekteris researched the non-contact optical device for a two-dimensional straightness measurement of a machine and presented the measuring principle and measuring transversal displacements of machine parts in two directions during their linear longitudinal motion [13]. Wang presented a simple and light-weight two-dimensional straightness measurement configuration based on optical knife-edge sensing and analyzed the physical model of the configuration and performed simulations. The experimental results indicated that the configuration could achieve ±0.25 µm within a ±40 µm measurement range along a 40 mm primary axial motion [14]. Zhang proposed a three-degrees-of-freedom measurement system based on the Faraday effect for simultaneously measuring two-dimensional straightness errors and their positions and analyzed the influence of angle error of the semitransparent mirror on the straightness measurement [15]. Wang proposed a novel method for testing the straightness error of a long guideway using a laser tracker and researched the basic principle and testing process, whose testing precision was 0.4 µm/m [16]. Because of airflow disturbances, there were significant errors in terms of optical interference for the straightness measurement if the guideway was too long. Liu proposed a method for noise attenuation of straightness measurements based on laser collimation, and the signal noise was reduced by about 90% [17]. One of the greatest challenges of long-distance measurement of straightness is beam drift. To reduce beam drift, Li applied a digital optical phase conjugation method, which could effectively decrease the beam drift [18]. Ji presented a compensation method for the sensor gain difference using a mixed sequential two-probe method, and the results demonstrated that the proposed compensation method was effective for the measurement of the straightness error with sub-micron accuracy [19]. Li proposed a new straightness splicing method using the stable characteristic angle between two adjacent sub-guide rails for a long guideway when measuring the straightness error using an interferometer [20]. Yun proposed a rail corrugation measurement method based on the coarse-to-refined data splicing method, which could acquire the wavebands of corrugation varying from 10 mm to 1 m [21].

In order to measure the straightness error of the long guideway in an ultra-precision grinder accurately, a splicing measurement method using a high-precision flat mirror and a displacement sensor is proposed in this paper, and the mathematical model for straightness data stitching is established. Through measurement experiments, the correctness and accuracy of the stitching model were verified. In order to further improve the machining accuracy of the component, the straightness error compensation model was established. Using the error compensation method, the flatness error of large-scale optics was grinded to 2.98 µm.

## 2. Splicing Measurement of Straightness Error

### 2.1. Principle of Splicing Measurement

The measurement process of the guideway straightness error using a high-precision flat mirror and displacement sensor is shown in Figure 1. The high-precision displacement sensor was fixed on the spindle and moved along the guideway with the slide table. The flat mirror was fixed on the machine table, and the displacement sensor took measurements on the surface of the mirror. If the flatness error of the mirror was much smaller than the straightness error of the guideway, the data acquired by the displacement sensor could be considered the straightness error. To accurately obtain the straightness error at different positions of the guideway, a synchronous data acquisition card was used to collect the signal output using the displacement sensor and the grating ruler reading head.

It is difficult to manufacture a large and high-precision flat mirror, so it is almost impossible to directly measure the straightness of a long guideway. In order to accurately measure the straightness error of a long guideway, a segmented measurement method of the straightness error at different positions of the guideway was proposed, and the straightness error of the whole guideway was calculated using a stitching algorithm based on coordinate transformation. The schematic diagram of the segmented measurement of straightness error is shown in Figure 2. Firstly, the flat mirror was fixed at position 1 (black solid line), and the displacement sensor was moved along the mirror’s surface from point A to point B. In this process, the straightness error within the AB range was acquired, which could be represented by a data series (*x_i_*, *y_i_*) in the measurement coordinate system *O*_1_*X*_1_*Y*_1_, where *i* = 1, 2, 3, …, *n*. Then, the flat mirror was fixed at position 2 (blue dashed line), with a certain overlap area CB between position 2 and position 1. To ensure the effectiveness of the measurement, the length of the CB area was generally greater than 15% of the total length of the flat mirror, and the displacement sensor was moved along the mirror surface from point C to point D. In addition, the straightness error within the CD range was acquired, which could be represented by a data series (*x_j_*, *y_j_*) in the measurement coordinate system *O*_2_*X*_2_*Y*_2_, where *j* = 1, 2, 3, …, *m*. Within the CB area, the straightness errors of the two measurements were consistent. As shown in Formula (1), after rotation and translation transformation, the straightness error (*x_j_*, *y_j_*) of the CB area measured with a flat mirror fixed at position 2 was consistent with the straightness error (*x_i_*, *y_i_*) of the CB area measured with a flat mirror fixed at position 1, where *i* = *k*, *k* + 1, *k* + 2, …, *n*, *j* = 1, 2, 3, …, *q*. Bringing the measured data series (*x_i_*, *y_i_*) and data series (*x_j_*, *y_j_*) into formula (1), the rotation angle *α* and translation vector (*a*, *b*) could be solved based on the least square method. After rotation transformation by angle *α* and translation transformation by vector (*a*, *b*), the straightness error (*x_j_*, *y_j_*) of the BD area was combined with the straightness error (*x_i_*, *y_i_*) of the AB area, where *i* = 1, 2, 3, …, *n*, *j* = *q* + 1, *q* + 2, *q* + 3, …, *m*, and the combined data was the straightness error of the full AD area.
(1)$$\left[\begin{array}{c} x_{i}\\ y_{i} \end{array}\right]=\left[\begin{array}{c} x_{jT}\\ y_{jT} \end{array}\right]=\left[\begin{array}{cc} \cos\alpha & \sin\alpha \\ -\sin\alpha & \cos\alpha \end{array}\right]\left[\begin{array}{c} x_{j}\\ y_{j} \end{array}\right]+\left[\begin{array}{c} a\\ b \end{array}\right]$$

### 2.2. Experiments on Measuring Straightness Error of Guideways

In order to verify the correctness of the splicing measurement method, experiments measuring the straightness error were carried out on an ultra-precision grinding machine with hydrostatic guideways and position closed-loop feedback control. As shown in Figure 3, the length of the *x*-axis, *z*-axis and *y*-axis were 1400 mm, 800 mm and 550 mm, respectively. Firstly, the straightness error of the *z*-axis at the *y*-axis direction was measured using a flat mirror (No. 1) with a size of 440 mm × 40 mm and flatness error of 0.08 µm (Figure 4) and a spectral confocal displacement sensor LECU-D27A23R06S23-UP with an accuracy of 60 nm, as shown in Figure 5. The flatness error of the mirror was much smaller than the straightness error of the guideway, so the influence on the measurement results could be ignored. The *z*-axis was longer than the length of the flat mirror. It was necessary to measure the straightness error of the *z*-axis guideway at two positions, which were −800~−360 mm and −440~0 mm in the machine coordinate system.

As shown in Figure 6, the straightness error of the *z*-axis at the −800 mm to −360 mm position was measured five times along the positive and negative directions, respectively, with a measuring velocity of 600 mm/min and data sampling period of 100 ms. When measuring along a single direction, the results of five measurements of straightness error were consistent. Subtracting the average of five measurements of straightness error from every set of data to evaluate the repeatability, as shown in Figure 7, the maximum deviation did not exceed 0.15 µm, which was far less than the straightness error. Using the same method, the straightness error at the −440~0 mm position was measured, as shown in Figure 8.

According to the above stitching processing algorithm, the straightness errors measured at the −800~−360 mm position and −440~0 mm position were processed, as shown in Figure 9. Then, the straightness error of the whole *z*-axis was directly measured using the No. 2 flat mirror with a size of 840 mm × 40 mm and a flatness error of 0.12 µm, as shown in Figure 10. The straightness error curve measured using the splicing method was the same as the straightness error curve directly measured. As shown in Figure 11, the maximum difference of the straightness error between two measurement methods was 0.3 µm, which was far less than the straightness error of the guideway 5.8 µm. Therefore, the segmented measurement and data splicing processing algorithm could be used for precise straightness error measurements of the long guideway.

The straightness error of the *x*-axis was measured using the No. 2 flat mirror. The length of the *x*-axis was 1400 mm, but that of the flat mirror was 840 mm. It was necessary to measure the straightness error of the *x*-axis guideway at two positions, which were −1400~−560 mm and −840~0 mm in the machine coordinate system. After the segmental measurement and data splitting process, the straightness error of the whole *x*-axis was calculated, whose PV was 5.5 µm, as shown in Figure 12.

## 3. Compensation for Straightness Error of Guideways

### 3.1. Analysis of Straightness Error Compensation

In the grinding process, the machined surface was formed according to the distance between the component and the grinding wheel [22]. The straightness errors of the feed axes would be 1:1 copied on the component surface and would reduce machining accuracy [23]. To improve processing accuracy, Liang modeled the geometric accuracy of the on-machine measurement system and compensated for the geometric error in the optical freeform surface machining machine [24]. There are high-precision grating rulers for position measurement and closed-loop feedback control in the ultra-precision machine, and the positioning errors of the feed axes are usually much smaller than the straightness errors. Therefore, it is possible to compensate for the straightness errors of the *x*-axis and *z*-axis by correcting the position of the *y*-axis during processing [25], which can improve the machining accuracy of the component. In the compensation grinding process of the flat optical element, because the straightness curves are not monotonic, the grinding wheel will move up and down with the distribution of the straightness errors, and the reverse motion error of the *y*-axis will reduce the compensation effect. To prevent the influence of the reverse motion error on component accuracy, a novel straightness error compensation method, as shown in Figure 13, was proposed. Firstly, the straightness error curve would be rotated to be monotonic. Then, according to Formula (2), the error matrix *E*(*i*, *j*) of the whole machining area was calculated, where *e_X_*(*i*) was the rotated *x*-axis straightness data, and *e_Z_*(*j*) was the rotated *z*-axis straightness data. Finally, according to the component positions in the machine coordinate, the error matrix *E_S_*(*i*, *j*) within the component area was cut out from error matrix *E*(*i*, *j*), which was used to 1:1 calculate the relative positions of the grinding wheel and the component in the grinding process. After grinding, the straightness error should be compensated for.
(2)$$E(i,j)=e_{X}(i)+e_{Z}(j)$$

### 3.2. Compensation Grinding Experiment

As shown in Figure 14, the straightness error compensation grinding experiment of the flat fused quartz optical element with a size of 1400 mm × 500 mm was carried out on the ultra-precision grinder, whose positioning error of the *y*-axis was 1.2 µm measured using a laser interferometer. The grinding parameters are shown in Table 1. According to the *x*-axis straightness error and *z*-axis straightness error, as shown in Figure 9 and Figure 12, the error matrix of the component area was calculated using Formula (2), as shown in Figure 15. The flatness of the optics after grinding without error compensation was measured using the above confocal displacement sensor, and the P-V value was 7.54 µm, as shown in Figure 16. After compensation grinding, the P-V value of the flatness error was reduced to 2.98 µm, as shown in Figure 17, which was less than the straightness errors of the guideways. The straightness error of the *x*-axis and *z*-axis of the grinder had been well compensated for.

## 4. Conclusions

In this paper, a segmented measurement method for the straightness error of a long-distance guideway in an ultra-precision grinder was proposed, as well as an error data splicing algorithm based on coordinate transformation. Comparative measurement experiments of the guideway straightness errors were carried out on an ultra-precision grinder. The maximum difference between the straightness errors of the splicing measurement and direct measurement was 0.3 µm, which was far smaller than the guideway straightness error of 5.8 µm. It was accurate enough to measure the guideway straightness error using the segmented measurement and data splicing algorithm. To reduce the influence of the guideway straightness error on machining accuracy, the straightness error compensating method based on error curve rotation and *y*-axis position correction was proposed. Finally, the compensation grinding experiment of a flat fused quartz optical element with a size of 1400 mm × 500 mm was carried out. Without error compensation grinding, the P-V value of the flatness error of the element was 7.54 µm. After error compensation grinding, the P-V value of the flatness error was 2.98 µm, which was less than the straightness errors of the guideways. The experimental results indicated that the compensation method had a suppressive effect on the guideway straightness errors.

## Figures and Tables

**Figure 1 micromachines-14-01670-f001:**
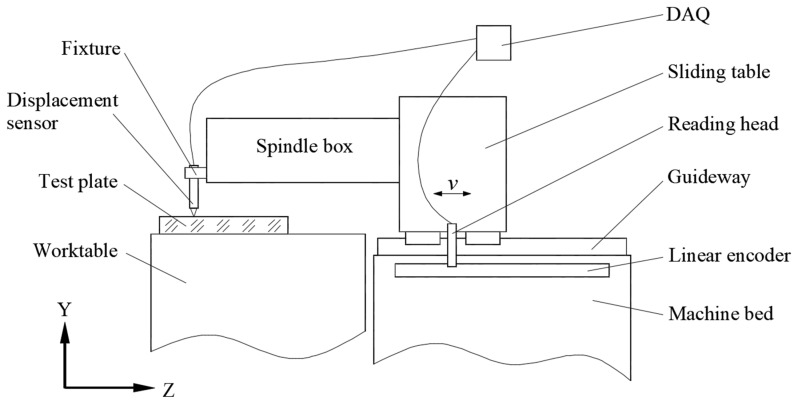
Principle of measuring straightness with a flat mirror and displacement sensor.

**Figure 2 micromachines-14-01670-f002:**
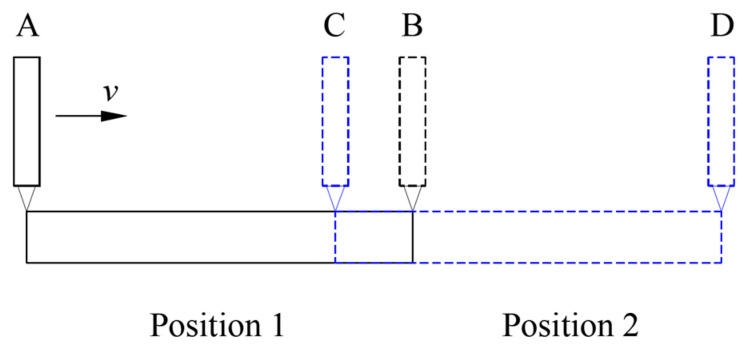
Schematic diagram of segmented measurement.

**Figure 3 micromachines-14-01670-f003:**
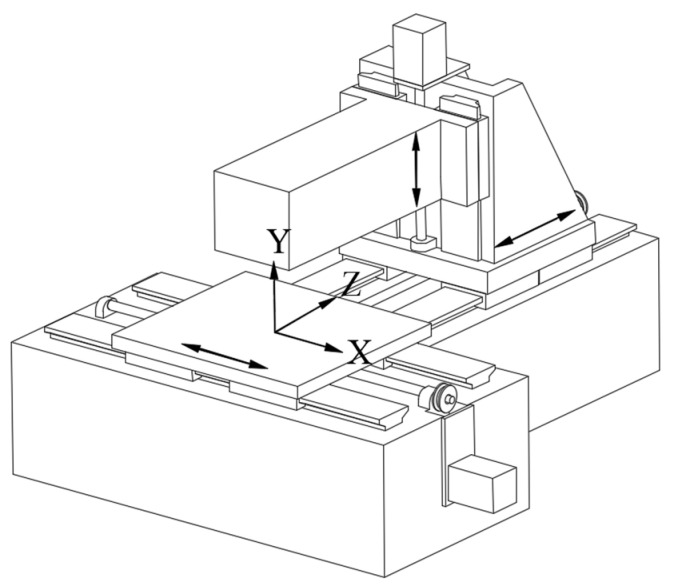
The three linear axes of the ultra-precision grinding machine.

**Figure 4 micromachines-14-01670-f004:**
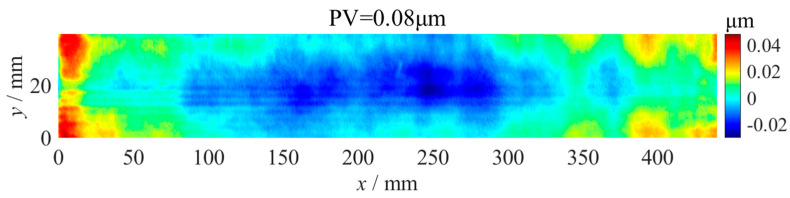
Flatness error of the No. 1 flat mirror.

**Figure 5 micromachines-14-01670-f005:**
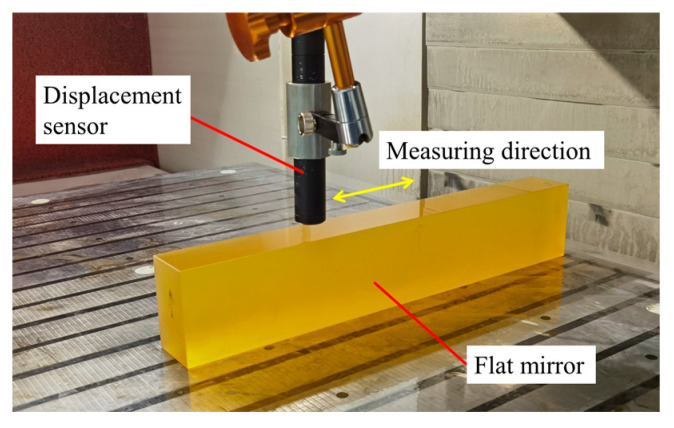
Measuring process of the straightness error.

**Figure 6 micromachines-14-01670-f006:**
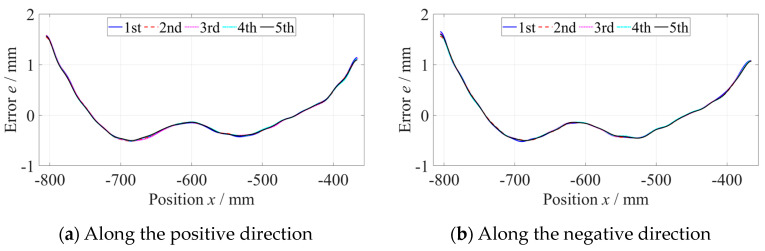
Straightness error of the *z*-axis at the −800~−360 mm position.

**Figure 7 micromachines-14-01670-f007:**
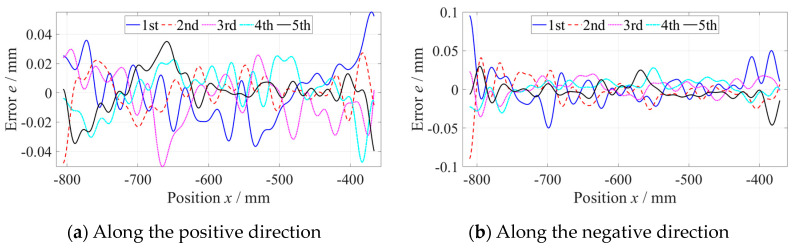
Repeatability standard deviation of the straightness error.

**Figure 8 micromachines-14-01670-f008:**
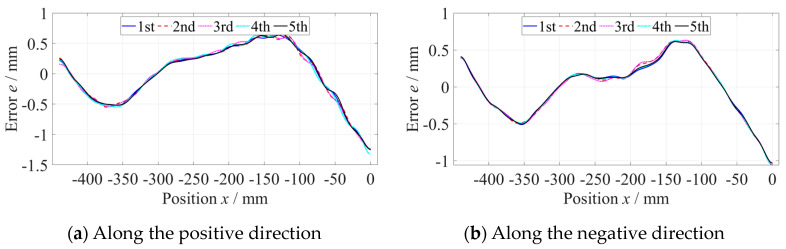
Straightness error of the *z*-axis at the −440~0 mm position.

**Figure 9 micromachines-14-01670-f009:**
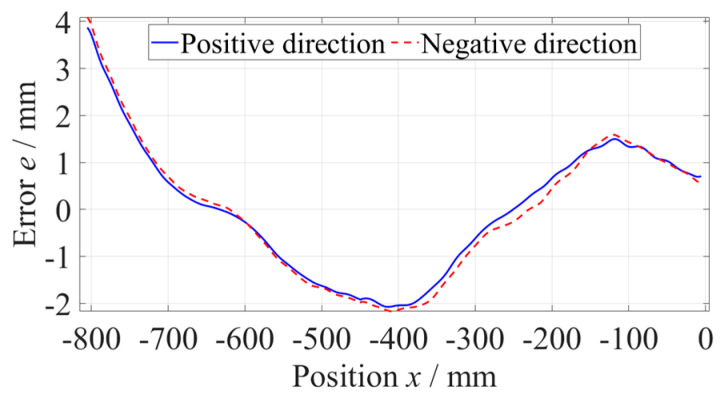
Straightness error after data splicing.

**Figure 10 micromachines-14-01670-f010:**
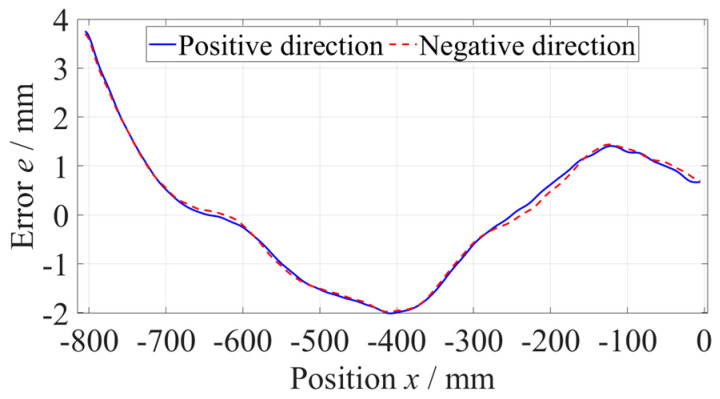
Straightness error measured using the No. 2 flat mirror.

**Figure 11 micromachines-14-01670-f011:**
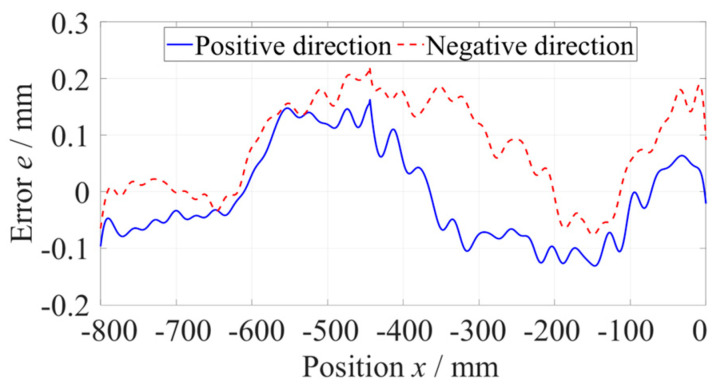
Difference between the stitching measurement and direct measurement.

**Figure 12 micromachines-14-01670-f012:**
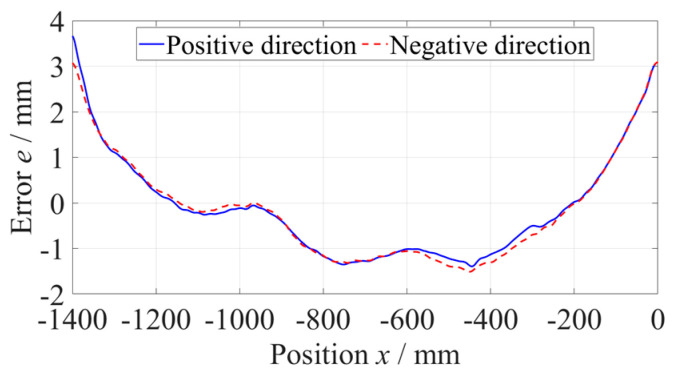
Straightness error of the *x*-axis.

**Figure 13 micromachines-14-01670-f013:**
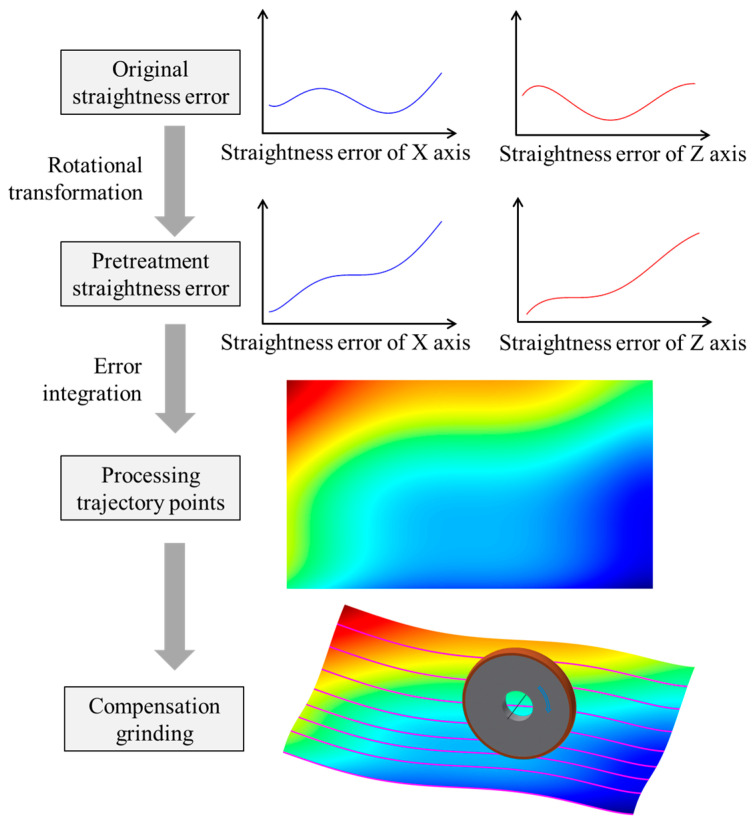
Principle of straightness error compensation.

**Figure 14 micromachines-14-01670-f014:**
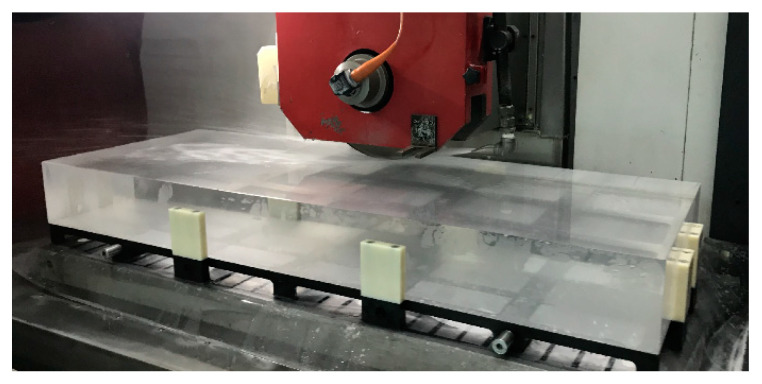
Grinding experiment of flat optics.

**Figure 15 micromachines-14-01670-f015:**
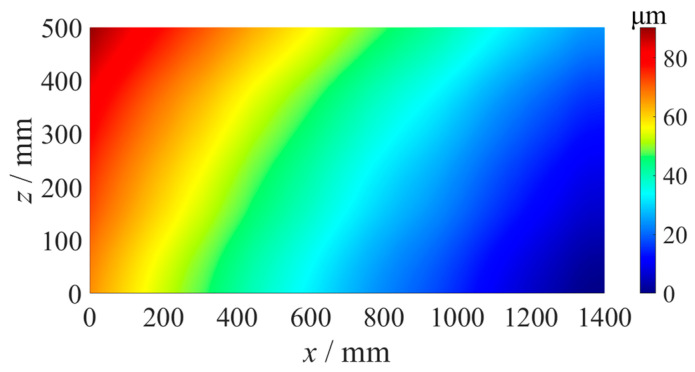
Grinding wheel motion trajectory points.

**Figure 16 micromachines-14-01670-f016:**
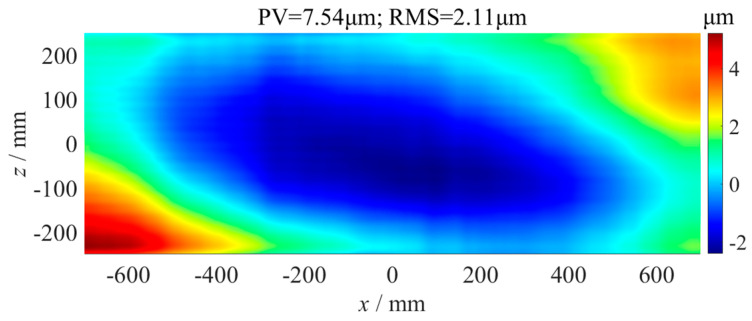
Flatness error of optics after grinding without error compensation.

**Figure 17 micromachines-14-01670-f017:**
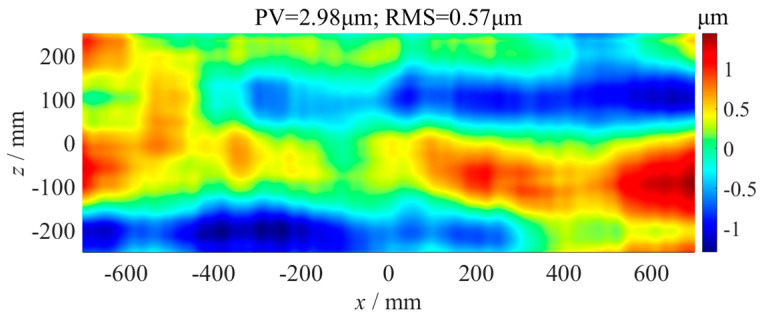
Flatness error of optics after error compensation grinding.

**Table 1 micromachines-14-01670-t001:** Grinding parameters.

Grinding Wheel	Coolant	Grinding Velocity	Feed Speed	Grinding Depth per Time
Diamond wheel with a grit size of 8~12 µm	Water	30 m/s	5000 mm/min	10 µm

## Data Availability

No new data had been created.

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
