# Peer review of "Splicing Measurement and Compensation of Straightness Errors for Ultra-Precision Guideways"

_micromachines, 2023, doi:10.3390/mi14091670_

Round 1

Reviewer 1 Report

1.  The quality of the figures can be improved to make it more clear;

2. Most of the time, the paper just describes the method to compensate the error, but the scientific problems and difficulty should be analyzed and discussed;

3. The authors should compare their method with the previously published one in manufacturing free-form surface.

There are some grammatical errors that should be corrected.

Author Response

Dear Reviewer,

Thank you very much for your review of this article. In response to your review comments, I have made modifications to the original text, which are explained as follows:

  1. The quality of the figures can be improved to make it more clear;

Response: All unclear figures such as Figure 4, Figure 6 and so on have been modified to be clear.

  1. Most of the time, the paper just describes the method to compensate the error, but the scientific problems and difficulty should be analyzed and discussed;

Response: The scientific problem is calculating the error matrix of the whole machining area according to the guideway straightness error. Due to that the straightness curves are not monotonic, the grinding wheel will move up and down with the distribution of straightness errors, and the reverse motion error of the Y-axis will reduce the compensation effect. So the difficulty is eliminating the influence of Y-axis reverse error on the compensation process by rotating the straightness error to be monotonic.

  1. The authors should compare their method with the previously published one in manufacturing free-form surface.

Response: The compensation method of geometric static error in free-form surface manufacturing has been compared.

  1. There are some grammatical errors that should be corrected.

Response: The entire text has been carefully checked again. The expression and grammatical errors have been corrected.

Reviewer 2 Report

1. The innovation points of the paper are general. Please further elaborate on the innovation points, and explain the complete large-scale linear guide error compensation method in the literature review section.

2. Please add a coordinate system in Figure 1 for a better intuitive understanding of the publicity 1

3. Please mark the model of the displacement sensor, the accuracy of the displacement sensor, and whether it meets the sub-micron profile requirements

4. Please add some NC codes, data before compensation and after compensation

5. Do you consider the installation error between the plane mirror and the worktable?

The language  needs to be further improved, please check carefully

Author Response

Dear Reviewer,

Thank you very much for your review of this article. In response to your review comments, I have made modifications to the original text, which are explained as follows:

  1. The innovation points of the paper are general. Please further elaborate on the innovation points, and explain the complete large-scale linear guide error compensation method in the literature review section.

Response: The general compensation method of large-scale linear guide error is inputting the straightness error into the CNC system of the machine during the equipment debugging stage, as shown in Reference 6. This is an effective method for compensating straightness errors, but the CNC system requires additional sag error compensation function, which greatly increased the cost of CNC systems. On CNC machine tools without sag error compensation function, the straightness error compensation method proposed in this article can be used to directly iterate the straightness error of the guideways to the machining trajectory, achieving deterministic compensation for straightness error.

  1. Please add a coordinate system in Figure 1 for a better intuitive understanding of the publicity 1

Response: The coordinate system has been added in Figure 1.

  1. Please mark the model of the displacement sensor, the accuracy of the displacement sensor, and whether it meets the sub-micron profile requirements

Response: The model of displacement sensor is LECU-D27A23R06S23-UP and the accuracy is 60nm, which has been supplemented in page 4.

  1. Please add some NC codes, data before compensation and after compensation

Response: The flatness data of element before compensation and after compensation have been added in the article.

  1. Do you consider the installation error between the plane mirror and the worktable?

Response: The installation error between the flat mirror and the workbench will introduce tilt error into the final data obtained. During data processing, the tilt errors will be decoupled, and do not affect straightness error data.

  1. Comments on the Quality of English Language: The language needs to be further improved, please check carefully.

Response: The entire text has been carefully reviewed again and expression errors have been corrected.

Reviewer 3 Report

Authors propose a simple splicing method to be applied to the measurement of the straightness error on long guideways in ultra-precision grinders.

In order to assess the straightness displacement (herein refereed as straightness error) a high-precision flat mirror and a precision displacement sensor are used.

The splicing procedure is describe. Lines 105/106 "Then the flat mirror was fixed at position 2, with a certain overlap area between position 2 and position 1". How the overlap is chosen or calculated?

The quantitative results are presented in a metrologicaly incorrect way - no uncertainety is indicated or described (the standard deviation of the straightness error repeatability is, however, presented at figure 7 - line 145 "According to formula 2, the repeatability standard deviation curve for 5 measurements was calculated and shown in Figure 7"). In fact terms like "about" (lines 20/21) "The maximum difference between the two measurements was about 0.3μm, which was far less than the straightness error of 5.8μm.") or "approximate" (line 161 for instance "...straightness error between two measurement methods was approximately 0.3μm..." are used instead throughout the text. Please correct that.

The effectiveness of the straightness error compensation algorithm to suppress the influence of straightness error on machining accuracy is not sufficiently proved and demonstrated and this should be done in order for the paper to be suitable for publication.

I suggest to extent the bibliography search too.

No major issues

Author Response

Dear Reviewer,

Thank you very much for your review of this article. In response to your review comments, I have made modifications to the original text, which are explained as follows:

  1. The splicing procedure is described. Lines 105/106 "Then the flat mirror was fixed at position 2, with a certain overlap area between position 2 and position 1". How the overlap is chosen or calculated?

Response: To ensure the effectiveness of the measurement, the length of the CB area was generally greater than 15% of the total length of the flat mirror. This information has been added in the article.

  1. The quantitative results are presented in a metrologicaly incorrect way - no uncertainety is indicated or described (the standard deviation of the straightness error repeatability is, however, presented at figure 7 - line 145 "According to formula 2, the repeatability standard deviation curve for 5 measurements was calculated and shown in Figure 7").

Response: What needs to be expressed here is that it has high repeatability for multiple measurements. In the revised version, the average of five measurements of straightness errors is subtracted from every data to evaluate the repeatability. The maximum deviation was not exceeding 0.15µm, which was far less than the straightness error. That means the measurement method has high repeatability.

  1. In fact terms like "about" (lines 20/21) "The maximum difference between the two measurements was about 0.3μm, which was far less than the straightness error of 5.8μm.") or "approximate" (line 161 for instance "...straightness error between two measurement methods was approximately 0.3μm..." are used instead throughout the text. Please correct that.

Response: Expressions of uncertainty like those have been modified.

  1. The effectiveness of the straightness error compensation algorithm to suppress the influence of straightness error on machining accuracy is not sufficiently proved and demonstrated and this should be done in order for the paper to be suitable for publication.

Response: In the revised version, the flatness data of element before compensation and after compensation have been added. The flatness of optics after grinding without error compensation was more than 7µm. After compensation grinding, the P-V value of flatness error was reduced to 2.98µm. The straightness error of the X-axis and Z-axis of the grinder had been well compensated.

  1. I suggest to extent the bibliography search too.

Response: The bibliography search has been extended in the revised version.

Reviewer 4 Report

1. In this paper, a segmented measurement method for straightness error of long distance guideway in ultra-precision grinder was proposed, as well as error data splicing algorithm based on coordinate transformation.

2. The topic is original and relevant in this field. This eliminates some gaps in this area.

3. The straightness error of guideway is one of the key indicators of ultra-precision machine, which plays an important role in the machining accuracy of work-piece. In order to measure straightness error of long distance ultra-precision guideway accurately, a splicing measurement for straightness error of guideway using high-precision flat mirror and displacement sensor was proposed in this paper, and the data splicing processing algorithm based on coordinate transformation was studied. Then comparative experiments on splicing measurement and direct measurement of straightness error were carried out on a hydrostatic guideway grinder.

4. Specific improvements in methodology and additional control measures are not required.

5. The conclusions are consistent with the evidence presented.

6. Links are appropriate.

7. Additional comments to tables and figures are not required.

The work has theoretical and practical significance.

Author Response

Dear Reviewer,

Thank you very much for your review and confirmation of the research content of this article. The entire text has been revised and improved again.

Round 2

Reviewer 1 Report

The authors have addressed the comments.

Reviewer 2 Report

The paper has been approved and revised according to the reviewer's comments.

The paper has been approved and revised according to the reviewer's comments.

Reviewer 3 Report

Authors try to improve their paper. Although the authors seems to have a limited knowledge about metrology, the work reported as some merits and deserves being published in current form